# FUME 2.0 - Flexible Universal processor for Modeling Emissions

Michal Belda[1], Nina Benešová[2], Jaroslav Resler[3], Peter Huszár[1], Ondřej Vlček[2], Pavel Krč[3], Jan Karlický[1], Pavel Juruš[3], and Kryštof Eben[3]

[1]Department of Atmospheric Physics, Faculty of Mathematics and Physics, Charles University, Prague, V Holešovičkách 2, 18000, Prague 8, Czech Republic
[2]Czech Hydrometeorological Institue, Na Šabatce 2050/17, 143 00 Prague 12, Czech Republic
[3]Institute of Computer Science, Czech Academy of Sciences, Pod Vodárenskou věží 271/2, Prague 8, Czech Republic

**Correspondence:** Michal Belda (michal.belda@matfyz.cuni.cz)

**Abstract.** This manuscript introduces FUME 2.0, an open-source emission processor for air quality modeling, documenting the software structure, capabilities, and sample usage. FUME provides a customizable framework for emission preparation tailored to user needs. It is designed to work with heterogeneous emission inventory data, unify it into a common structure, and generate model-ready emissions for various chemical transport models (CTMs). Key features include flexibility in input data formats, support for spatial and temporal disaggregation, chemical speciation, and integration of external models like MEGAN. FUME employs a modular Python interface and PostgreSQL/PostGIS backend for efficient data handling. The workflow comprises data import, geographical transformation, chemical and temporal disaggregation, and output generation steps. Outputs for mesoscale CTMs CMAQ, CAMx, WRF-Chem, and large-eddy simulation model PALM are implemented along with a generic NetCDF format. Benchmark runs are discussed on a typical configuration with cascading domains, with import and preprocessing times scaling near-linearly with grid size. FUME facilitates air quality modeling from continental to regional and urban scales by enabling effective processing of diverse inventory datasets.

## 1 Introduction

Air quality chemistry transport models (CTMs) are nowadays widely used for many purposes: air quality forecasting, regular air quality assessment, assessment of the impact of emission scenarios, or source apportionment to mention just a few (e.g. Colette et al., 2012; Solazzo et al., 2013; Im et al., 2015a; Syrakov et al., 2016; Im et al., 2015b; Marécal et al., 2015; Ďoubalová et al., 2020; Coelho et al., 2022). CTMs need different input data, where meteorology and emissions are among the most important ones. While the preparation of meteorological inputs in a real application is quite straightforward (typically hourly meteorology on a 3D grid is taken from a numerical weather prediction or regional climate model), preparing emissions may be challenging, especially when emissions from different sources need to be combined. The input files usually come in various formats, units, geometries, and resolutions. The emissions need to be spatially disaggregated to the model grid, and some of the basic pollutants (typically total particulate matter, volatile organic compounds and nitrogen oxides) need to be chemically speciated and annual emission totals need to be split into hourly data. Some of this processing is very likely region- or country-specific, which complicates the task even more. This is why software for emission processing needs to be flexible enough to

minimize manual preparation of the data as much as possible, it also needs to be able to process effectively large amounts of

25 data and deliver them in a reasonable time. Such an emission processor is a complex tool, which is not easy to design and develop. Our group started the development of a new emission processor in 2014, when only a limited number of open-source emission processors was available. One of the widely used was SMOKE (v3.6 at that time; UNC, 2017), which was originally designed for the U.S. and its application for other regions requested substantial adaptations. Some examples are its application for Spain (Borge et al., 2008), SMOKE for Europe (Bieser et al., 2011), or SMOKE-Asia (Woo et al., 2012). Another limitation

of SMOKE at that time was the need for manual emission preprocessing, especially spatial disaggregation of emissions, which represents the major amount of workload.

Since then, a number of standalone and integrated emission processors appeared in the modeling community. Within the COSMO modeling framework, an online emission module has been developed (Jähn et al., 2020). On the other side of the spectrum are several standalone emission models, e.g. High-Elective Resolution Modelling Emission System (HER-

35 MES; Guevara et al., 2019), Emission Inventory Processing System (EMIPS; Chen et al., 2023) or Air Emission Processor (https://github.com/ctessum/aep).

Out of these, HERMES is probably the most actively developed and best described, however, it has some limitations: e.g. in HERMESv3_GR, users can only choose global and regional inventories contained in the HERMES library. While this can be convenient and time-saving in many applications, it poses severe limitations when specific local emission inventories need to be

used. Here we introduce the FUME processor attempting to overcome the limitations of these existing emissions preprocessors. FUME gives users more flexibility in some areas, such as the selection of emission inventories, output projection definition, or spatial disaggregation. FUME also allows to use spatial proxies (e.g. floor area of houses heated by certain types of fuel, or area of arable land) during the transformation of emissions to the output grid, which makes this process much more flexible than simple regridding of the original data. Another feature that makes FUME probably unique is the possibility to prepare

emission inputs not only for meso- but also for microscale CTMs (currently represented by LES model PALM) taking into account specifics of the urban environment, e.g. multiple levels of surfaces with emissions.

In this manuscript, FUME is described in the most recent version, 2.0, as publicly available at https://github.com/FUME-dev/fume.

## 2   Description of FUME

FUME stands for *Flexible Universal processor for Modeling Emissions*. It is open-source software created and maintained by a team of researchers from the Czech Academy of Sciences, Czech Hydrometeorological Institute, Charles University, and Czech Technical University. It was designed to process a variety of emission data in multiple file formats together with chemical speciation and time disaggregation coefficients and generate emission files directly ingestible by various popular CTMs (including 3D emissions). In this regard, the processor was designed to be extensible to allow adding support for more

input and output formats in the future. Other main features of the FUME design are independence on input data, applicability at different scales (local/regional/continental), transferability between locations, and support for various geographical systems.

The processor implements the widely accepted disaggregation model for emission flows:

$$E(p,l,t,s) = \sum_i \sum_c Z(i,r,c) \cdot q_p(i,p) \cdot q_l(r,l,c) \cdot q_t(t,c) \cdot q_c(r,s,c) \cdot q_s(r,c),$$

where

$E(p,l,t,s)$ ... output emission flow for a polygon $p$ (typically a grid cell), vertical level $l$, time $t$ and output species $s$

$Z(i,r,c)$ ... primary emission of emitted species $r$ from source $i$ and category $c$

$q_p(i,p)$ ... spatial disaggregation coefficient from source $i$ into polygon $p$

$q_l(r,l,c)$ ... vertical disaggregation coefficient for emitted species $r$ from category $c$ into vertical level $l$

$q_t(t,c)$ ... time disaggregation coefficient for category $c$ into time $t$

$q_c(r,s,c)$ ... chemical speciation coefficient for emitted species $r$ and for category $c$ into output species $s$

$q_s(r,c)$ ... emission scenario coefficient for emitted species $r$ and for category $c$

Individual parts of this model are described in Sect. 2.2 and 2.3 and in the user guide.

The software implementation of FUME is based on freely available open-source software packages and libraries. Most of its components are platform-independent, although it is primarily intended to be run on Linux-based HPC systems. However, FUME also supports external modules which could be platform-specific. The processor is built upon three main software packages (optional external modules may include additional requirements):

- Python v3.6 or higher – the processor user interface is written in Python 3. Some FUME modules require a small number of additional freely available libraries (see the documentation for the current list of all dependencies).

- PostgreSQL v10 or higher (tested on 10.23) – The PostgreSQL relational database management system (RDBMS) serves as a data storage backend.

- PostGIS v2.4 or higher – the GIS extension of PostgreSQL enables a variety of GIS operations on geospatial data.

All data ingested by FUME get internally converted to a relational database in the PostgreSQL/PostGIS RDBMS (*backend*). With its rigid structure, the RDBMS ensures that a unified and consistent set of input data is available before processing begins. The unified inner structure allows flexibility in input/output data format support and easy extensibility within the Python interface (e.g. in case the input emission data files are in a currently unsupported format or a support needs to be added for additional CTM). The computational core is implemented as configurable backend-executed transformation chains enabling to process different source groups individually.

The user interface (*frontend*) is represented by a command-line utility `fume` written in Python. The frontend has a modular structure so that it is easy to replace current modules with different ones or add new modules. Among others, this allows interfacing with external models, several of which are already implemented (e.g. biogenic emission model MEGAN). Each processor run is controlled by a *workflow* file specifying which modules are to be executed and in which order. All user settings

can be specified in the FUME configuration described in the user documentation with example configuration files provided as well.

The processor workflow can specify any number of Python functions to be run in the order they are listed in the file. The workflow usually consists of these main processor modules: static data setup, emission and support data import, initialization of the case projection and grid, geographical transformation chains, application of emission scenarios, processing point sources, meteorology data collection, chemical speciation, preparation of time disaggregation coefficients, external model calls, and export to CTM input files. The user may choose to skip any of these steps; a typical context for this is changing the output grid for the same input data, in which case the import module does not have to be executed again. This way, computational requirements can be significantly reduced.

A typical processor workflow, dependencies of the components, and data flows among them are shown in Fig. 1 and described in detail further in this chapter.

## 2.1 Import and unification of the input data

The main part of the import process is the import of emission sources – information about their location, emission flux, or activity data (density or count, e.g. cattle/cow population, vehicle density, number of heating units, etc.). Besides the emission sources, supplementary geometries (e.g. for masking of the data or application of surrogates) can also be imported in this process. The data can come from different sources and thus are highly heterogeneous in format (shapefile, text file, NetCDF, Excel, etc.), coordinate system, geometry (point, line, polygon, grid), species naming conventions, units, etc. The import module is configurable and designed to deal with various inputs so that the data preparation requested from the user is as low as possible. As of writing this manuscript, text (e.g. CSV), shapefile and NetCDF data formats are supported for point, area, and line sources, and thus, FUME can deal with standard emissions sources such as CAMS (Kuenen et al., 2022), EMEP (Wankmueller, 2019) or EDGAR (Crippa et al., 2018).

Emission sources are internally flagged 'P', 'A', or 'L' (point, area, line) for convenience. In principle, this distinction is not necessary for the geometrical processing within the database as the information about the source type is expected to be implicitly available through the geometrical metadata, and thus PostGIS procedures have all the geometrical information about the sources (regardless of this flag) necessary for the gridding stage. However, the flag can be used when the user needs to process different types of sources differently, either during the geometrical transformation stages or for output. A typical use case is a CTM that expects emissions in separate files for line/area and point sources.

Aside from emission data, the user must also provide other inputs as information about the temporal distribution and chemical speciation of emissions (see section 2.3), emission sector nomenclature in the form of hierarchical category definition, inventory and model species list, etc. Those are provided as simple text files with a predefined structure. Further, optional text-based inputs can be provided to define emissions scenarios and the vertical distribution of the emissions. The configuration files in these cases are a direct replacement for traditionally hard-coded features such as chemical mechanisms (in the SMOKE-compatible gspro files) or emission categories and allow users to customize or add more of these without changing the processor code. Example files for commonly used CTMs are provided for user convenience in the tutorial cases.

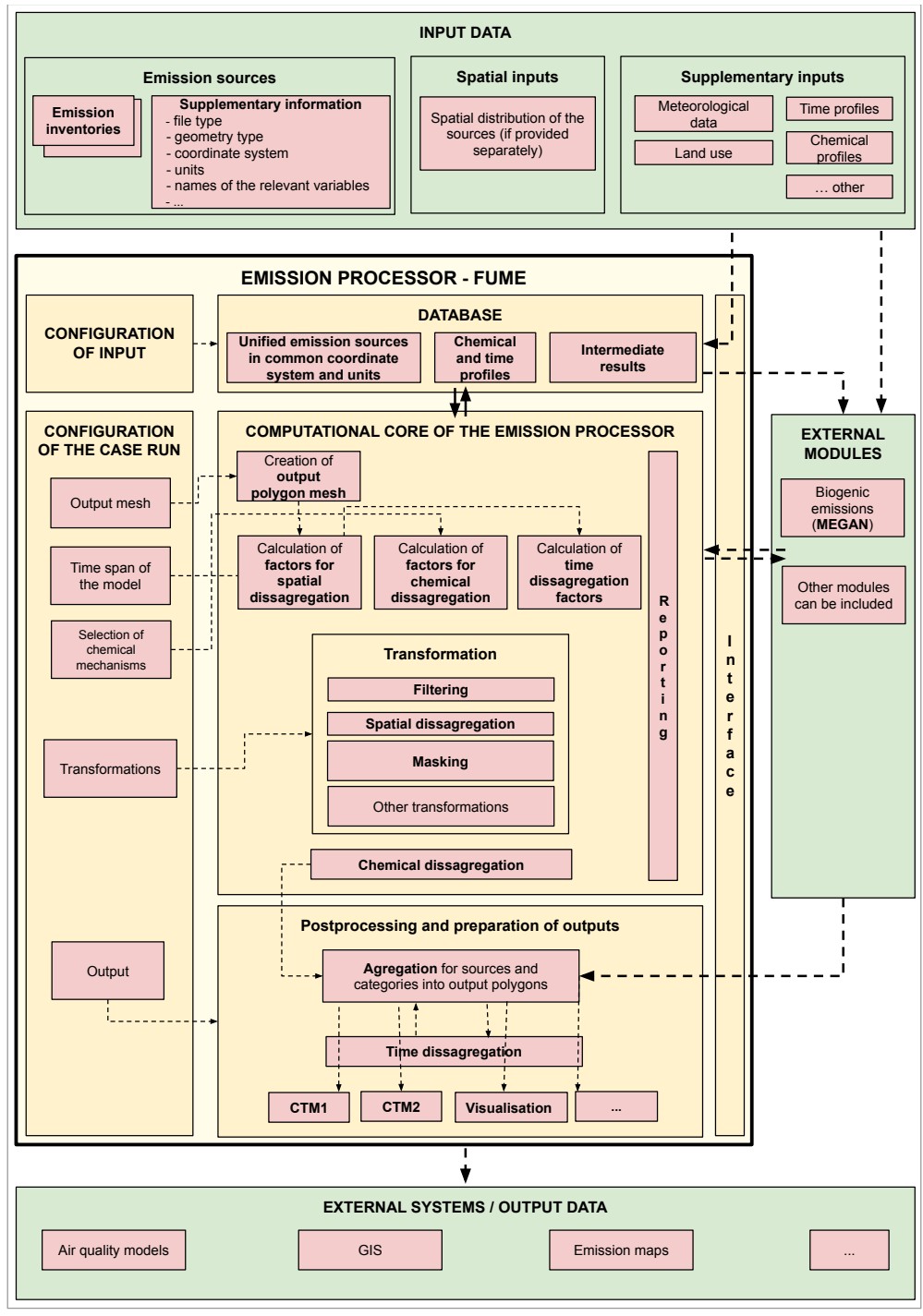

**Figure 1.** FUME workflow

Meteorological data can be required for running some external models, e.g. MEGAN for biogenic emissions (Guenther et al., 2012). These data are usually taken from numerical weather prediction or regional climate models. The current version contains import modules for the WRF (Weather Research and Forecast; Skamarock et al., 2019) model (NetCDF format) and

the ALADIN (Aire Limitée, Adaptation Dynamique, Development INternational; Termonia et al., 2018) model (GRIB format). New import interfaces for other models can be easily implemented by using existing input modules as templates.

All input data (emission sources and additional inputs) are preprocessed, the emissions are stored as the so-called emission sets based on user configuration (this makes it possible to deal with emission sets separately in different transformation chains) and transformed into a unified structure during the import process. This way, the consecutive computational steps are indepen-

dent of the original data format. This is in accordance with the modular philosophy of the software and allows to easily extend the import layer to deal with other data formats without the need to change the other parts of the software.

## 2.2   Transformations

The FUME computational core is implemented as a configurable set of transformations of the emissions from the source inventories and sets to the model grid. The transformations can be configured in chains that run independently of each other

and in the final stage produce numerical coefficients mapping the source geometry to the output geometry. The main built-in transformations are thus ones that perform geometrical operations (intersection, masking, surrogates). For further flexibility, supporting transformations are included for filtering operations (selection by emission inventory, emission set, source type, and vertical level) and applying emissions scenarios. Chaining these transformations then allows for defining common processes, e.g. applying a scenario on emissions inside/outside a country or region from an emission inventory based on predefined

polygons.

After processing all transformation chains, the transformation process is finalized by collecting all geometrical coefficients that map source shapes to the output grid. The coefficients are then applied to the original emission data producing gridded emission totals for the configured case. The current version supports output to regular grids in commonly used geographical projections (supported by the PostGIS backend). A modification is in development to allow exporting emissions on irregular

grids and arbitrarily defined polygons.

## 2.3   Chemical and temporal disaggregation, vertical distribution

The emission data coming from the emission inventories are typically specified for a different set of species expected by a specific chemical mechanism. To accommodate for this discrepancy, the processor needs to perform chemical speciation, i.e. the process of translating between the two sets of species (e.g. split aggregated VOC emissions from emission inventories into

separate species expected by the CTM, like in the CB05 mechanism). The speciation factors are user-supplied and the format is consistent with the GSPRO file format used by the emission processor SMOKE (UNC, 2017).

The temporal disaggregation is performed in two steps. In this part of the workflow, multiplicative factors for each output time step are pre-calculated and stored in the database. The time disaggregation itself (i.e. calculation of the time series) is not performed in this step but rather at the final step - the output to final format for performance reasons. Thus, by default, emission

time series are not stored in the database. The temporal factors are supplied by the user as daily/weekly/monthly factors and/or explicit hourly factors (time series) for the period of interest.

The speciation and time disaggregation coefficients can be specified for each emission category or hierarchy of categories individually and must be supplied by the user.

Moreover, FUME implements three main techniques of vertically distributing the emissions. In the first approach, users can

define emission category-dependent vertical distribution factors to arbitrary levels (given by geometric height above ground). Each inventory/emission set can be assigned its vertical distributions (if not defined, emissions from the particular invento- ry/emissions set are considered as surface emissions). The application of vertical distribution factors is performed later in the *postprocessing* step (see below), i.e. after the speciation and after calculating the time disaggregation factors. In the other approach used currently for PALM 3D emission, users can assign a vertical level representing the order of the surface from

top to bottom in case of full 3D geometry terrain allowing to distribute emissions e.g. over and under bridges or overhang- ing buildings. This assignment is done in the transformation stage separately for each chain. These levels are then applied in the *postprocessing* output module by matching them to corresponding vertical layers in the PALM grid based on terrain information read from the PALM static driver.

Technical description of the level assignment procedure is available in the user guide. The third approach is used for point

sources. Point source inventory can carry additional information like stack height and diameter or temperature and velocity of the emitted species. These attributes can be used for the vertical distribution of the emission in the *postprocessing* output module (currently PALM module) or transformed into specific point sources emission input of the particular CTM (currently implemented e.g. in CAMx or CMAQ output modules).

## 2.4  External models

For additional emission calculations, FUME supports the inclusion of third-party models. All models are implemented using a unified interface which ensures the independence of the subsequent steps. This allows for adding new modules or changing existing ones without the need to modify the rest of the code. The presented version of the FUME includes full support for the MEGAN biogenic emission model (version 2.1; Guenther et al., 2012). Additionally, FUME contains two further modules for emission from heating and emissions of ammonia ($NH_3$) from agriculture, both depending on actual meteorological conditions.

These are however experimental modules, currently under development and their usage is advised to be consulted with the authors.

It has to be noted that external models and the emission fluxes they calculate usually depend on meteorological conditions (e.g. biogenic emissions by MEGAN) which are read in by FUME from regional scale weather prediction or climate models (see section 2.1 for details). In principle, the data can be provided on any projection and spatial/temporal resolution and

FUME internally interpolates the data in space and time, provided that the input data cover the FUME domain and time frame. In situations when emissions exported from FUME are used in a CTM model, FUME technically allows to use different meteorological data (e.g. from a different model) than in the CTM run. However, it is recommended to use the same driving data for consistency.

## 2.5 Postprocessing

The computational core of the processor computes spatially and chemically disaggregated data with precalculated temporal factors. Those data are ready for further use in external software, primarily CTMs or in some visualization software. In FUME, this process is handled by the built-in *postproc* module. The design of the module is modular to allow for easy implementation of submodules for other chemical models without repeatedly accessing the internal database structure. All processor outputs are internally realized by a data distribution bus that collects all output requirements, reads required data from the database once, and distributes them among all output submodules. Reading data from the database backend is performed by a *provider* object - an object that acquires the data in packages and distributes them among registered *receivers* specified by the user in the processor configuration. Currently, the *postproc* module supports output for the regional CTMs CMAQ (https://www.epa.gov/cmaq), CAMx (https://www.camx.com/), WRF-Chem (https://ruc.noaa.gov/wrf/wrf-chem/ and urban microscale model PALM (http://palm-model.org/). CMAQ version 5.0 and higher are supported, i.e. the area sources are stored as NetCDF 2D files while point sources are in the form of text files of the defined format. Output for CAMx also supports both area and point sources in a Fortran binary file format based on an original UAM-IV standard (EPA, 1990). The PALM submodule supports output of the 2D PALM emission files as well as the output of the full 3D emissions to the generic volume source file according to the PALM Input Data Standard (PIDS; Heldens et al., 2020). All NetCDF-based output formats are derived from a generic NetCDF module that can be used as a template for the implementation of other postprocessing modules.

The database backend is designed to work efficiently with large amounts of data. However, generating time series of emissions from the database into a format usable by CTMs can be quite time-consuming, especially in cases with large domains spanning multiple time zones, using large emission data sets and/or very detailed category sets. The *postproc* module was thus designed in a two-step form. In the first step, an intermediate NetCDF file is saved containing annual emission sums for each model species and category distributed over the selected domain. If vertical distribution factors were defined by the user, the emissions are redistributed vertically during this step (in one of the ways described in section 2.3) and written in the intermediate file. Then in the second step, multiplication by the pre-generated time factors (see section 2.3 for details) with the emissions from the intermediate file is performed resulting in emission time series. This second step is performed outside of the database which generally results in much faster processing. This approach is advantageous in tasks such as daily operational runs when the bulk of the processing (spatial and chemical) can be performed only once and the time-specific output then efficiently generated on the fly from the intermediate file. The two-step technique is used as a default in the implementation of all currently available output submodules.

If the need arises for the support of other chemical models or output formats, it can be easily implemented thanks to unified interfaces and the modular character of the output module. Besides the CTMs output formats, the processor also allows simple plotting using the `matplotlib` graphical library. The plots are implemented as maps of the spatial distribution of emission totals for individual chemical species. The visualization is implemented mainly for quick checking of the gridded emissions. Detailed diagnostic checks are provided by the *reporting* module described in the 2.6 section.

## 2.6  Reporting

Runtime diagnostics is supported by the *reporting* module. It is designed to perform continuous tracking of the emission data and various checks during their processing. It records all the input files, together with information such as recognized categories and species, scenarios applied, vertical distribution, etc. Further, the software performs checks to detect possible errors in the input data, e.g. whether all sources are assigned chemical and time disaggregation factors or checks whether the time factors sum up to 1. Finally, the reporting module provides checksums during and at the end of the process. The sums are annual emission totals for each emission set and species. This is greatly useful to check that no emissions are double-counted or no emissions are missing. The reporting outputs are currently implemented as human-readable text files.

## 3  Example usage

### 3.1  Illustration of FUME outputs

The capabilities of FUME have been thoroughly tested in various applications from the meso-scale to the meter-scale context of LES simulations. In this section, we provide examples of complex processing chains and the outputs in a visual form. Figure 2 shows emissions generated by FUME in a testing setup used for the benchmark discussed in detail in sect. 4. Three levels of zoom are shown corresponding to three nested domains in horizontal resolutions going from 0.1° to 0.04° for the intermediate and to 0.005° for the inner-most domain.

The second example presents FUME-generated emissions for validation simulations with the LES model PALM for Prague, Czech Republic. Figure 3 shows NO emissions in a traffic-heavy region of the Prague center. Emissions are shown as vertical column totals in 2m horizontal resolution for the full computational domain (a) and a detailed look at a smaller area (c) and a vertical cross-section (d). Vertical distribution of traffic emissions from cars and trains (ground), residential heating (roofs) and point sources (isolated elevated dots) is clearly visible in (d).

### 3.2  Tutorial user cases

For learning purposes, data for two test cases are available along with the FUME source code showing most of the processor's capabilities while also providing suitable configuration templates. The first test case shows a simple operational forecast configuration for a simulation in Europe based on a combination of regional-scale emission data from CAMS/EMEP databases and local emissions for the Czech Republic. It features all the main processor modules: input data import, geographical transformations, application of simple scenarios, speciation and temporal disaggregation, and output into CAMx and CMAQ area emission files.

The second case focuses on a micro-scale workflow for the preparation of emission inputs for the LES model PALM. It presents a typical usage of the transformations, preparation of 3D emissions, and utilization of the PALM output module for producing emission files according to the PALM Input Data Standard (PIDS). The case represents a downsized version of a larger domain shown in sect. 3.1 with a selected subset of emission inventories. It is based on simulations of a traffic-heavy

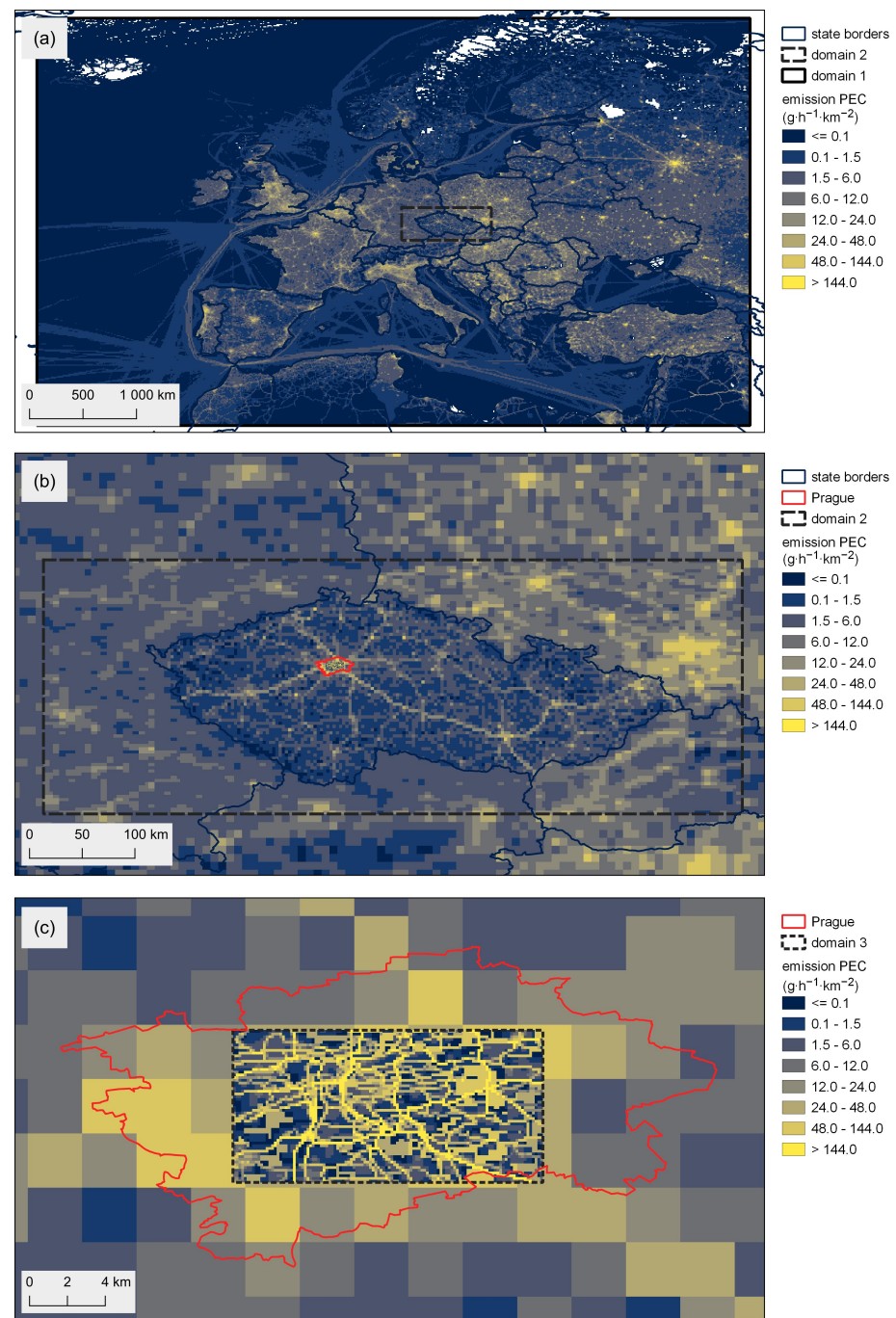

**Figure 2.** FUME outputs for the three testing domains, elemental carbon, 1st June 2019 9 AM. Emission fluxes are normalized to 1 × 1 km$^2$. Administrative boundaries © EuroGeographics.

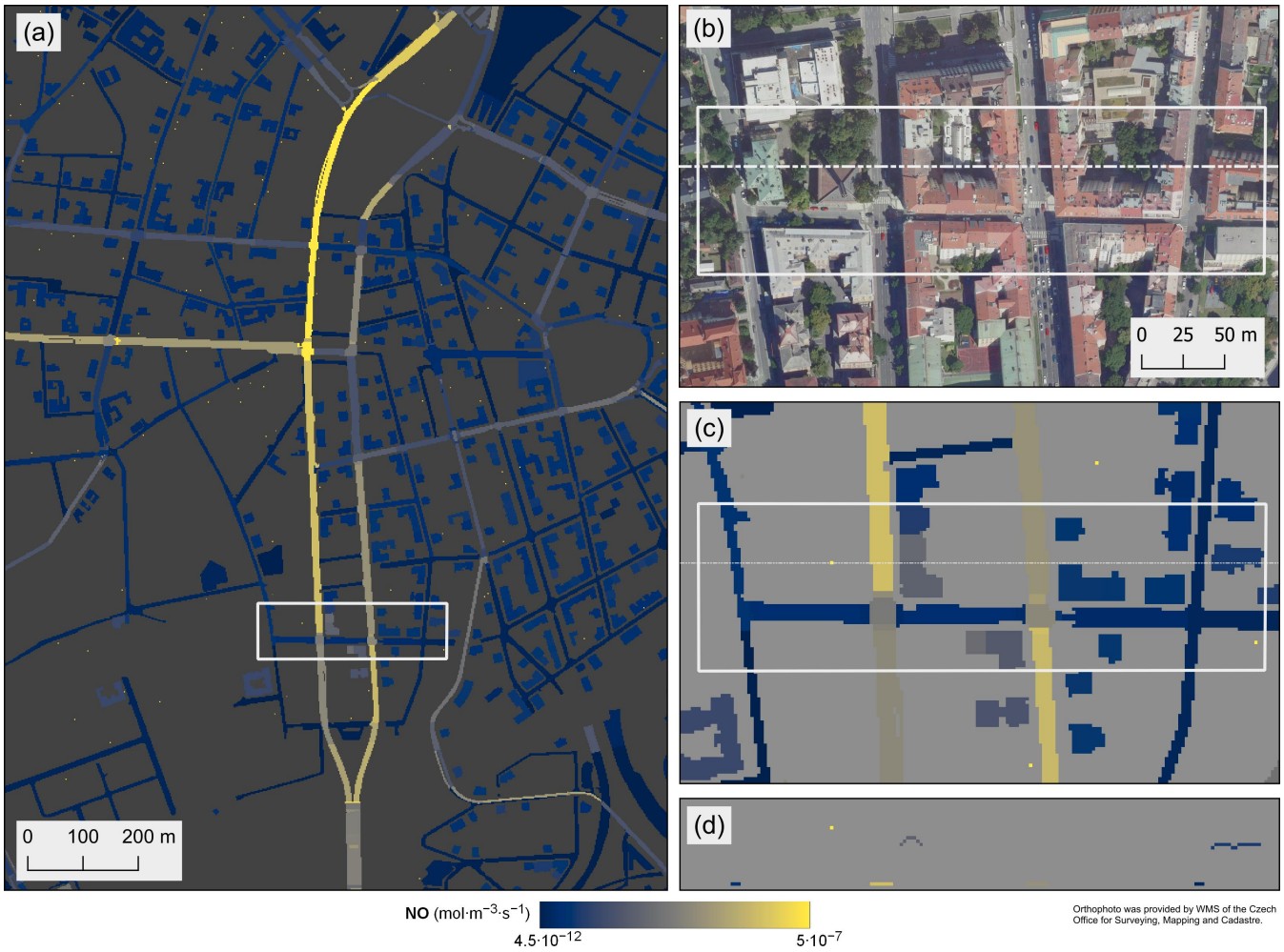

**Figure 3.** Emission of nitrogen monoxide (NO) for the PALM model in $\mathrm{mol/m^3/s}$ on 13 February 2023 at 6:00 AM. The panels (a) and (c) show NO vertical column totals. The panel (b) shows an aerial photograph corresponding to the area shown in the panel (c). The panel (d) is the vertical cross-section along the dashed line in panels (b) and (c). White rectangles in panels (a), (b) and (c) denote the same area.

area in Prague, Czech Republic. It shows a complex case combining regional-scale area emissions for residential heating disaggregated via detailed building description with high-resolution traffic emissions transformed via spatial surrogates. These data are transformed and processed into 2D surface and 3D volume-source emission files for PALM.

## 4   Computational efficiency & benchmark

To show the performance of the FUME we set up 3 nested domains. The first (d01) replicates the CAMS European air quality forecast domain (Marécal et al., 2015). The horizontal coverage is: west boundary = 25.0° W, east boundary = 45.0° E, south

boundary = 30.0° N and north boundary = 70.0° N while the horizontal resolution is 0.1° x 0.1° (700 x 400 grids). The second domain (d02) covers the Czech Republic and ranges from 10.8° E to 19.6° E in west-east direction and from 48.2° N to 51.4° N in south-north direction. The horizontal resolution is 0.04° x 0.04° (220 x 80 grids). The third domain (d03) covers the center part of the capital city Prague and ranges from 14.18° E to 14.75° E and from 49.92° N to 50.20° N with horizontal resolution 0.005° x 0.005° (114 x 56 grids).

Emissions included in this case:

- European inventory - CAMS data CAMS-REG-AP_EUR_0.05x0.1_anthro_v4.2-ry_yearly_2019 (Kuenen et al., 2022). This includes 4055822 area (grid) and 20622 point sources defined in a 490 MB text file.

- Czech emissions (data from the national Register of Emissions and Air Pollution Sources REZZO; detailed road transport emissions based on the traffic census of 2016 were prepared by ATEM (Ateliér ekologických modelů - Studio of ecological models)http://www.atem.cz):

    - point sources (2019) - 45143 point sources provided as a 10 MB text file.

    - local heating (2019) - 106485 area (grid) sources, given as $10 \times 77$ MB text files (one file per fuel type). The geometry information is defined in an accompanying shapefile.

    - quarries, coking plants, foundries (2019) - 622 area sources given as a 1 MB polygon shapefile.

    - road traffic sources (2016) - 149608 line sources given as 87 MB shapefile.

The list does not constitute all available Czech emissions. Those data were chosen to cover different types and formats of input data.

The import process of the above emission data takes about 140 minutes. It is important to keep in mind that this process needs to be performed only once. Then the data are in the unified inner FUME format ready for further processing.

Further, the emissions need to be prepared for the particular case i.e. mainly emission gridding for the particular domain and also other calculations as chemical speciations for chosen chemical mechanism and the NetCDF intermediate file creation. The time demands of this process depend on the size of the domain and the number of sources included. For the presented case the transformation and speciation process takes about 260 minutes and the intermediate NetCDF file creation takes 180 mins for the d01 domain, 2 minutes for transformations and 12 minutes for NetCDF creation for the d02 domain and 38 s transformations and 90 s NetCDF creation for the d03 domain. Again it should be emphasized that unless the domain grid or emission sources included are changed, this process is done only once. The size of the intermediate NetCDF files are 519 GB, 195 MB and 3 MB for d01, d02 and d03 domains, respectively.

The last step is the final preparation of output files for the selected date. For this example, we chose the 1$^{st}$ June 2019. To produce one day i.e. 24 hours, it takes 90 minutes for d01, 3 minutes for d02 and 1 minute for d03 domain. The times are summarized in the Tab. 1. The presented case comprises around 1000 emission categories (for d01), due to country-specific chemical speciation factors, which markedly increases the time and disk space demands. The computational demands could

**Table 1.** Times needed to process data in the benchmark case.

|  | d01 | d02 | d03 |
|---|---|---|---|
| case preparation | 260 mins. | 2 mins. | 38 s |
| intermediate file creation | 180 mins. | 12 mins. | 90 s |
| final 1-day output | 90 mins. | 3 mins. | 60 s |

be significantly lower in the cases where speciations are not region-specific. The example output for a selected hour for three domains and species elemental carbon is in Fig. 2.

All database schemas take up 37 GB of disk space, of which 12 GB is the storage of the emission inventory import data and 15 GB is taken by the schema with all intermediate tables created for the processing of the d01 case.

To show the scaling computational time depending on the number of cells and domain extent we performed additional computations. As a reference calculation, we used the benchmark for d01 domain with CAMS emissions only. Then we calculated six additional runs while gradually decreasing the number of cells with the same resolution thus shrinking the domain extent. The computational costs were thus reduced not only by the lower number of grid cells but also by the lower number of emission sources and categories contained in the domain (categories in our benchmark are country-specific). The resulting dependencies for three main processes (case preparation, intermediate file creation and final 1-day output) are in Fig. 4. The case preparation calculation time increases approximately linearly with the increasing number of cells. The other two processes — creation of intermediate output file and final 1-day output — deal with writing to the storage and have stronger time dependency. The performance of the output module can, however, be highly influenced by the hardware and software configuration of the computer. For instance, writing large files to an external storage (e.g. NFS accessed network storage) can severely impede the speed of writing the files, especially for large domains.

The software produces several reports to monitor the emission processing by FUME. In particular, to ascertain the correctness of the whole process the sums of total emissions divided by species and emission sets are produced. This can be used to check that the total emission sum remains the same throughout the process. An example of this file for the d01 domain can be found in the supplement.

## 5 Conclusions

The Flexible Universal processor for Modeling Emissions (FUME) version 2.0 was introduced in this text. It described FUME's most important features such as the import of the raw emission inventories, the spatial and temporal disaggregation of the emission fluxes, speciation of the emitted species groups, and the postprocessing of data which includes output into standard CTM-ready formats or visualizing them as figures. The flexibility in support for various raw input formats possible as well as for arbitrary emissions categories and species was stressed. An example of its implementation for a cascade of European domains and central European subdomains using European and Czech emission inventories was provided as a benchmark along

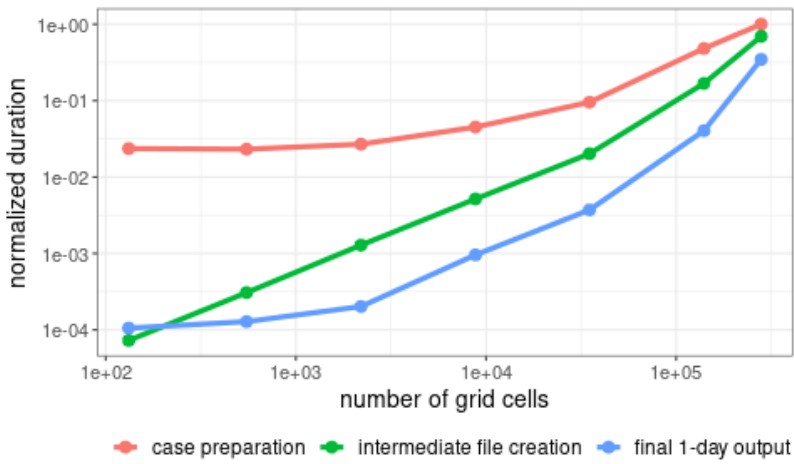

**Figure 4.** Scaling FUME calculations in dependence on grid cell number. Both axes are log scale and y values are normalized to 1 for the maximum value.

with the scaling for different grid sizes. The capability to include external models for calculating specific emissions (such as biogenic emissions) was also described.

Among the strengths of the FUME, we can name that it is free and open source software. It is highly configurable and flexible in terms of the format of input emission sources. It allows advanced source processing — filtering, scenario application, etc. It currently supports output for three air quality models. Throughout the development process, consideration has been given to the possibility of software extensibility. This makes it easy to add support for further input or output formats or incorporate specific emission calculation modules. In some cases, a design choice was made not to define hard-coded data flows, e.g. for overlaying regional inventories. The authors opted instead for a more flexible approach of user-configurable transformation chains that allow for any combination of inventories, categories, and species.

FUME is under active development and a number of ideas for future development were suggested to further extend its functionality and flexibility. Some of the areas that have been identified for improvements are geographically dependent speciation and temporal factors, output to arbitrarily defined polygons (i.e. not only regular grids), more flexible configuration files, and helper utilities for generating commonly used configuration groups. In some cases, we encountered issues with performance,

especially when writing outputs for large computational domains, so one of the focuses of the development team is to improve computational speed. With larger domains and datasets, some optimization of the hardware and software configuration of the computer can provide much performance improvement (e.g. exporting emissions to local storage). Finally, improvements are planned to provide better user support in terms of preparing recommendations for configurations for different situations and more detailed software documentation for developers.

FUME has already been proven to be useful in many studies focusing on air quality in central Europe. It has been used for the annual air quality assessment in the Czech Republic and was used for the 2018 and 2023 update of the Air Quality Plans for the Czech Republic (https://www.mzp.cz/en/air_quality). FUME is also used in the operational high-resolution air-quality forecasting system "Libuše" (http://libuse.urbipragensi.cz/aq/) created for Prague within the URBI PRAGENSI project (Ďoubalová et al., 2020) and was also applied in emission calculation for assessing the impact of urban emissions over central Europe (Huszar et al., 2021, 2022, 2024), the impact of natural emission like wind-blown dust (Liaskoni et al., 2023) as well as the impact of the urban canopy meteorological forcing on both gas-phase and PM air-quality (Huszar et al., 2018b; Huszár et al., 2018a; Huszar et al., 2020b, a). Recently, it has been used for emission calculation also in a regional long-term ozone study (Karlicky et al., 2024).

Regarding the preparation of emissions for the microscale model PALM, FUME has been used for the processing of urban emissions having high spatial and time resolution inside the projects TURBAN (TURBAN, 2023b) and (ARAMIS, 2023b). This complex task involves the processing of a number of different types of emission inputs, utilization of different spatial surrogates including emission levels for the creation of 3D emission, other transformations, and writing outputs to 3D PALM volume sources emission file according to the PALM PIDS format.

FUME proved to be a crucial tool for combining the diverse forms of different emission data used in these studies (from European-level regional emissions to national and urban scales). It also enabled effective scenario calculations or geography mask-dependent processing (e.g. when the urban and non-urban emissions had to be distinguished), and surrogate utilization. Its high flexibility makes it implementable in very diverse cases that air-quality modelers all around the world can potentially face.

*Code and data availability.* The FUME 2.0 code and user documentation can be obtained through https://zenodo.org/doi/10.5281/zenodo. 10017150 or directly at the public repository at https://github.com/FUME-dev/fume. The pre-configured user test cases including all input data necessary for replication can be downloaded at https://doi.org/10.48700/datst.bf6s2-5tq48

*Author contributions.* MB, NB, JR, PH, PK, PJ, JK contributed to the development of the FUME's code and manuscript text, NB performed the FUME benchmarks presented in the manuscript, NB and JR created the user cases, OV, JK and KE contributed to testing the code and writing the manuscript.

*Competing interests.* No competing interests are present.

*Acknowledgements.* The development of FUME was financed by the following projects: grant of the Technology Agency of the Czech Republic (TA ČR) No. TA04020797 "Advanced emission processor utilizing new data sources" (TA ČR, 2023), LIFE-IP MAŁOPOLSKA project No. LIFE14 IPE/PL/000021 "Małopolska Region - Implementation of air quality plan for Małopolska Region - Małopolska in a healthy atmosphere", URBI PRAGENSI project financed by the Operational Programme Prague – Growth Pole of the Czech Republic No.
CZ.07.1.02/0.0/0.0/16_040/0000383 (URBI PRAGENSI, 2023), TA ČR project ARAMIS No. SS02030031 "Air Quality Research Assessment and Monitoring Integrated System" financed by the programme for applied research, experimental development and innovation in the field of environment - Environment for life (ARAMIS, 2023a; ARAMIS, 2023b), project TURBAN "Turbulent-resolving urban modeling of air quality and thermal comfort" financed by grant No. TO01000219 from Norway Grants and Technology Agency of the Czech Republic within the KAPPA Programme for applied research, experimental development and innovation ((TURBAN, 2023a) and (TURBAN, 2023b))
and the FOCI project No. 101056783 "Non-CO2 forcers and their climate, weather, air quality and health impacts" co-funded by European Union in the Horizon Europe Framework Programme.

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
