# Peer review of "FUME 2.0 - Flexible Universal processor for Modeling Emissions"

_EGUsphere, 2023_

## Author Response (AR1)

**Replies to the reviewer's comments**

**The authors wish to express their thanks to both the reviewers for their helpful comments and suggestions. In this document, we answer the questions, and the clarifications were included in the relevant parts of the manuscript in the revised version. Please note that to improve readability, some parts of the manuscript were reorganized, mainly the examples of usage were moved to a new section 3.**

**The reviewer's comments are in italics and the authors' replies are in bold.**

**Reviewer 1**

*Belda et al. present FUME 2.0 as a new emissions pre-processor that processes various emissions data for use in CTMs. The software is open-source and has a modular Python interface with a PostgreSQL/PostGIS database backend allowing for emissions to be aggregated into a database before generating output for various models.*

*The paper is clearly written and in scope for GMD. I recommend it for publication with a few minor questions on the capability of this tool.*

*1. The authors implement MEGAN as an "external model" for interfacing with FUME. Such "external models" generate emissions dependent on model state information (radiative flux, temperature, etc.) which is generally provided from met fields that are also provided to the CTM - so they usually run "online" within a model or emissions processor with the same met fields as the parent model. What met data is used as input to MEGAN when running with FUME, and how is consistency with the met fields used by the underlying CTM ensured?*

**The FUME's MEGAN module can currently handle NetCDF input files from WRF or RegCM regional weather prediction/climate models as well as grib data from the ALADIN model. In principle, these data can be provided on any projection and spatial/temporal resolution and FUME internally interpolates the data in space and time, provided that the input met domain and input files temporal coverage covers the FUME domain and time frame.**

**As for the consistency between the meteorological driving data for FUME (MEGAN) and that used for the CTM that takes MEGAN emissions, technically these can be entirely independent but of course, it is assumed and the best practice recommends to use the same driving data for both.**

**Mention of this possibility was added to the 2.4 section.**

*2. How are point sources stored in the system and later gridded for input to CTMs?*

**All sources are stored in the PostgreSQL/PostGIS database along with all necessary geometrical/geographical metadata. The sources are also internally flagged as 'P', 'A', or 'L' (point, area, line) for convenience. In principle, this distinction is not necessary for the geometrical processing within the database as the information about the source type is expected to be implicitly available through the geometrical metadata, and thus PostGIS procedures have all the geometrical information about the sources (regardless of this flag) necessary for the gridding stage. However, the flag can be used in situations in which the user**

needs to process different types of sources differently either during the geometrical transformation stages or for output. A typical use case is a CTM that expects emissions in separate files for line/area and point sources.

Clarification added in the 2.1 section.

*3. Figure 1. workflow mentions "Visualization" as an output. Does FUME provide any useful diagnostics for sanity-checks, e.g., annual emission totals for individual species, globally/per inventory/per country? This could be useful to check the results. As well as ensuring the totals are correct and no emissions are double-counted. It could be useful to show some of these visualization examples as well (is Figure 2 a direct output from FUME?)*

FUME contains a reporting module that is designed to provide such diagnostics, track the imported data, and check its processing. It records all the input files, together with information such as recognized categories and species, scenarios applied, vertical distribution, etc. Further, the software performs some checks to detect possible errors in the input data: it checks whether all sources are assigned chemical and time disaggregation factors or checks whether the time factors sum up to 1. Finally, the reporting module provides checksums during and at the end of the process. The sums are annual emission totals for each emission set and species. This is greatly useful to check that no emissions are double-counted or no emissions are missing. At this moment those reporting outputs are simple text files.

If demanded, FUME supports the output of simple emission plots using the Python matplotlib library. This allows users to plot either annual emission totals per species or plot for each time step and species. The output is configurable (file type, file name pattern, resolution, colormap, and if desired the plotter draws country borders or overlay user-supplied shp), however, it is meant to be used mainly for quick checks, not for the production of publication-ready figures. The internal interface of the postprocessing module allows easy implementation of other visualization routines if needed.

Information about the reporting module was added as the new 2.6 section.

*4. How does FUME aggregate, for example, a global inventory with regional inventories overlayed? Is there a systematic way that it will record and ensure no double-counting is applied? e.g. in HEMCO (Harmonized Emissions Component) a category/hierarchy system is used where emissions from different categories are summed together but only the highest hierarchy available is used.*

FUME does not use any implicit layer structure for inventories. The choice of which inventory to use at a particular polygon needs to be explicitly defined by the user in the transformation chains (e.g. use a global inventory over all Europe, but local REZZO inventory of the Czech Republic). This is by design, in our experiments, this proved to be a more flexible approach (e.g. for sensitivity studies). In the future development, we intend to implement a helper utility for generating configurations that for the time being must be defined explicitly.

Clarification was added to the Conclusions section.

*5. What kind of grids are supported for output?*

The version used in the manuscript supports any X/Y/Z regular grid on many geographical projections (any that the PostGIS can handle). However, a modification is in development that will allow to output the emissions on arbitrary polygons and irregular grids.

Clarification was added to the 2.2 section.

**Reviewer 2**

*The manuscript presents the technical description of the FUME emission model, together with a test case. The model has several advantages over previous existing emission processors and has been used widely in several studies. The manuscript is easy to follow and well-written. There are however a number of issues that needs to be addressed before the manuscript is suitable for publication.*

**Thank you for the comments and suggestions. The manuscript has been thoroughly revised, see our replies below.**

*General comments:*

*How about the vertical distribution of emissions?*

**Authors response:**

**FUME allows vertical distribution of gridded emissions using three different approaches described in the USER GUIDE Chapters 8.1 and 8.2.**

**The first approach performs vertical distribution using category-specific vertical distribution factors defined by the user for selected emission inventories or emissions sets along with the geometric height to where these emissions are to be distributed, while for one emissions inventory/set, multiple vertical distributions can be defined. FUME internally transforms the vertical distribution factors over the input inventory-specific heights to the user-defined heights and applies them once the original emissions are horizontally interpolated.**

**Another method was developed mainly for the purpose of placing emissions on the surfaces lying on top of each other. This situation occurs mainly in the case of fully 3D microscale simulations e.g. in the model PALM. An example of such emission can be e.g. street transport emission on a bridge over another street or railway with the street/railway transport emission. Another example can be a building with heating emissions overhanging the street.**

**A separate approach is used for point emission e.g. for stacks. FUME treats point emission in a special way which allows the inclusion of optional additional parameters such as stack height, diameter, gas temperature, and exit velocity. The height and other parameters can be used for the proper placement of the emission to the corresponding level in the output module. Depending on the particular target model, it can be done by 3D output emission (e.g. PALM 3D volume sources) or by the special point source emission file if the target model implements such an approach (e.g. CAMx and CMAQ).**

**In the revised manuscript, Section 2.3 was extended with a brief description of this feature. Later, in Section 2.5 we describe the application of the vertical distribution factors.**

*Which chemical mechanisms are supported?*

**In the design of the FUME, we tried to avoid any dependence on a particular target model or mechanism. The definition of necessary parameters of the mechanism (e.g. speciation to the target mechanism and model nomenclature) is part of the FUME configuration and can be easily extended by any other mechanism and model. For user convenience, the FUME chemical**

mechanism configuration file has a structure compatible with the SMOKE mechanism input (gspro file). The tutorial case distributed with FUME contains an example configuration for the CB6 mechanism and CAMx CF aerosol mechanism. The authors have used FUME with different mechanisms - e.g. CB05, CB6, SAPRC and aerosol mechanisms CAMx CF and CMAQ AERO and phstatp for PALM. The user guide contains a section dedicated to the question of how to create the configuration files from scratch if needed.

An explanation was added in the 2.1 and 2.3 sections.

*Which emission sectors nomenclatures are supported (SNAP etc.)?*

FUME does not rely on any specific nomenclature. The configuration of the nomenclature is a part of the case configuration. Input inventories can have their own nomenclature which is mapped into the common nomenclature of the FUME case during the import of the inventory. The internal inventory (in the example presented in the paper the GNFR nomenclature is used) uses a tree structure of the sectors, e.g. the sector-dependent configurations (speciation, time profiles) can be defined for any specific sector or a group of sectors. In the speciation and time disaggregation stage, FUME uses the specific sector configuration if it exists, otherwise falls back to the nearest existing ancestor definition in the tree. Detailed documentation on creating these configuration files is included in the user guide.

An explanation was added in the 2.1 section.

*How does FUME handle when the local proxy data and the input emission inventory (e.g. CAMS) do not match, e.g. missing proxies for some emission sectors in the local data?*

Similar to the chemical mechanisms and nomenclatures, FUME avoids hard-coded handling of data flows. A design choice was made instead to allow the user to define layering and other data flows in the form of configurable transformation chains. This allows e.g. to supply some sectors from detailed national-based data for the area of the country while sectors that are not available in the local data process from e.g. European-wide databases, e.g. CAMS, EMEP, etc. The transformation chains also include options to filter emissions based on source type (point, line, area), emission set (internally a subset of an emission inventory) and to use surrogates for spatial disaggregation. FUME does not attempt to provide any implicit handling of missing proxies, however, potential inconsistencies can be reported to the user by the reporting module.

A remark on this was added to the Conclusions section.

*Why is not a PALM case presented in the manuscript?*

An illustration of a real-life PALM case was added in the 3.1 section with examples of visualizations generated from FUME outputs.

As a second example, a brief description of a downsized PALM case in the form of a tutorial case was extended in the manuscript (now in the 3.2 section). The PALM case used for the tutorial shows a simplified version of the previous real-life case on a smaller domain with a selected subset of emission inventories (for computational feasibility reasons).

*Specific comments:*

*Line 110: Reanalyses met input (e.g. ERA5) is covered by the netcdf format or one needs to implement this as another input format?*

**The self-describing NetCDF format is generally supported by FUME, however, due to inconsistencies in metadata conventions (e.g. file, dimension and variable naming, units, etc.) used by individual meteorological or climate models, a specific import interface has to be implemented for each model separately or derived from an existing NetCDF interface (e.g. WRF).**

**A note on this was added to the corresponding paragraph in Section 2.1.**

*Line 157: How about WRF-Chem as mentioned in the beginning (Line 8)?*

**WRF-Chem postproc module is available in FUME. This information was added to the list of supported output formats in Abstract and sections 2.3 and 2.5.**

*Editorial comments:*

*Line 40: …. limitations OF these exisiting…..*

**Corrected.**